# Older Women Who Practiced Physical Exercises before the COVID-19 Pandemic Present Metabolic Alterations and Worsened Functional Physical Capacity after One Year of Social Isolation

**DOI:** 10.3390/healthcare10091736

**Published:** 2022-09-09

**Authors:** Kizzy Cezário, Carlos André Freitas dos Santos, Clineu de Mello Almada Filho, Gislene Rocha Amirato, Vitória da Paixão, Ewin Barbosa Almeida, Jônatas Bussador do Amaral, Adriana Caldo-Silva, Nuno Pimenta, António Rodrigues Sampaio, Pedro Teques, Fernanda Monteiro Rodrigues, Carolina Nunes França, André Luis Lacerda Bachi

**Affiliations:** 1Post-Graduation Program in Health Sciences, Santo Amaro University (UNISA), São Paulo 04829-300, Brazil; 2Discipline of Geriatrics and Gerontology, Department of Medicine, Paulista School of Medicine, Federal University of Sao Paulo (UNIFESP), São Paulo 04020-050, Brazil; 3Postgraduate Program in Translational Medicine, Department of Medicine, Paulista School of Medicine, Federal University of São Paulo (UNIFESP), São Paulo 04023-062, Brazil; 4Mane Garrincha Sports Education Center, Sports Department of the Municipality of Sao Paulo (SEME), São Paulo 04039-034, Brazil; 5ENT Research Lab., Department of Otorhinolaryngology-Head and Neck Surgery, Federal University of Sao Paulo (UNIFESP), São Paulo 04021-001, Brazil; 6Faculty of Sports Sciences and Physical Education—(FCDEF-UC), University of Coimbra, 3040-248 Coimbra, Portugal; 7Research Centre for Sport and Physical Activity, CIDAF-FCDEF-UC, 3040-248 Coimbra, Portugal; 8School of Social Sciences, Education and Sport, Polytechnic Institute of Maia, 4475-690 Maia, Portugal; 9N2i, Research Nucleus, Polytechnic Institute of Maia, 4475-690 Maia, Portugal; 10CIPER, Interdisciplinary Center for the Study of Human Performance, 1499-002 Lisbon, Portugal

**Keywords:** social isolation, lipid profile, protein profile, creatinine clearance, physical tests, aging

## Abstract

Background: Because the consequences of the lifestyle changes in older adults associated with the social isolation imposed in response to the COVID-19 pandemic are not fully understood, here, we investigated the effects of one year of social isolation imposed by COVID-19 on the metabolic parameters and functional physical capacity of older women who regularly practiced physical exercises before the pandemic. Methods: Systemic lipid and protein profiles, estimated creatinine clearance (ECC), and functional physical capacity (FPC) were assessed before (January-February 2020) and 12 months after social isolation in 30 older women (mean age 73.77 ± 6.22) who were engaged in a combined-exercise training program for at least 3 years before the COVID-19 pandemic. Results: In this group, we observed increased plasma levels of triglycerides and creatinine, an increase in the time necessary to perform gait speed and time-up-and-go tests, and reduced muscle strength assessed by the handgrip test and ECC post-COVID-19 pandemic relative to values recorded pre-pandemic. In addition, we observed significant correlations (both negative and positive) between anthropometric, some metabolic parameters, and physical tests. Conclusion: One year of interruption of physical exercise practice imposed in response to the COVID-19 pandemic significantly altered some systemic metabolic parameters and worsened ECC and FPC in older women.

## 1. Introduction

There is a consensus that the evident pace of older adult population growth can be considered one of the greatest victories of contemporary humanity, portraying the improvement in public health around the world. According to the World Health Organization (WHO), current projections are categorical in pointing out that, e.g., in 2050, the estimated population aged 60 years and older will be nearly 2.1 billion people [1].

In this prospective scenario, the development of chronic and degenerative diseases could also increase, and together with infectious and parasitic diseases, become the main mortality factors in older adults [2]. In this sense, it is widely accepted that a sedentary lifestyle, which is a common behavior observed in the aged population, increases the risk of the development of several chronic diseases and comorbidities, such as type 2 diabetes mellitus (T2DM), dyslipidemia, renal, and cardiovascular diseases (CVD), in addition to the decline in cognitive functions and functional physical capacities, which, in general, can lead to premature death [3,4].

The literature highlights that in order to achieve a healthy aging process, in which the person is able to perform basic daily functions with perfect and complete autonomy, it is necessary to present musculoskeletal and cardiovascular systems capable of performing daily activities without depending on third parties to perform actions they think are important. In this sense, the maintenance of physical activity, especially by performing exercise training, can assist in the preservation of not only physical but also mental and cognitive functions, making older adults capable of maintaining their independence for a longer period of time, in addition to mitigating the development and progression of chronic diseases and comorbidities [5,6,7]. Corroborating these data, our group has reported that older women who regularly practiced physical exercises showed improvements in several physical, metabolic, immunological, and redox indices parameters as compared to groups of inactive or sedentary older women [8,9,10]. Furthermore, we also demonstrated that the long-standing regular practice of combined-exercise training can maintain functional physical capacity in the older adult population [11], maintaining their capacity to perform the activities of daily living, which, according to the literature, can improve physical and mental health, socioeconomic independence, and social interactions in this population [12].

Regarding the assessment of functional physical capacities, gait speed (GS) and time-up-and-go tests (TUGT), in association with muscle strength evaluation by handgrip (HG) test, are not only broadly used in routine clinical and scientific studies but also are considered powerful tools to define the occurrence and progression of some aging diseases, particularly sarcopenia and frailty syndrome [13,14]. Moreover, anthropometric and systemic metabolic profile assessments are also crucial both to defining the clinical status of the individual and to assist in follow-up of the development and progression of chronic diseases and comorbidities [15]. In this sense, the results obtained in some systemic assessments, such as circulating profiles of lipid (total cholesterol and fractions) and protein (total protein and albumin), are routinely used to elucidate possible metabolic disturbances [16,17,18]. Particularly in terms of muscle evaluations, the systemic variation in the levels of urea and creatinine, as well as creatinine clearance, are widely used and can provide important information concerning musculoskeletal status, consequently helping to define the presence of muscular disorders, especially the occurrence of protein catabolism [19]. Beyond of this context, the analysis of serum creatinine levels and creatinine clearance is widely used to assess renal functions, especially in aged individuals [20]. Significant associations have been demonstrated with several metabolic parameters and physical tests, which can contribute to improved understanding of the aging process and guide the adoption of certain interventions, clinical or otherwise, to maintain functional physical capacity and achieve healthy aging [21,22].

Given that a sedentary lifestyle is closely associated with worsening of several clinical and physical outcomes and that respiratory infections are among the main causes of deaths in older adults, the current pandemic originated by the new coronavirus, SARS-CoV-2, facing humanity since March 2020 raises concern, especially in older adult subjects; for example, SARS-CoV-2 infection has been associated with an increased number of deaths in the aged population during this period [23,24]. Besides infection, additional concerns have been raised regarding the consequences associated with certain interventions applied by governments around the world in response to the pandemic, particularly social isolation, which led to the interruption of some healthy habits, e.g., the regular practice of exercise training programs [25,26].

Therefore, in the present study, we aimed to investigate the impacts of one year of social isolation imposed in response to coronavirus disease 2019 (COVID-19) caused by the SARS-CoV-2 virus on anthropometric and systemic metabolic parameters, such as lipid and protein profiles, as well as functional physical capacities; we evaluated these impacts GS, TUGT, and HG tests, as well as their associations, in a group of older women who regularly practiced a combined-exercise training program before the COVID-19 pandemic.

## 2. Materials and Methods

### 2.1. Participants and Design of the Study

This was a prospective, open-ended study with a blind analysis of outcomes, in which anthropometric data, systemic metabolic parameters, physical tests, and muscle strength were evaluated in a group of older women who practiced physical exercises pre-pandemic period (n = 30) on two occasions: before (pre) and 12 months after (post) the social isolation imposed in response to the COVID-19 pandemic. The number of older women volunteers enrolled in the present study was established according to sample calculation using the G ∗ Power program based on our previous study [8], with a confidence level of 95%.

Recruitment was performed in the Primary Health Care Program for Older Adult Individuals of the Geriatrics and Gerontology Discipline of the Federal University of São Paulo (UNIFESP), including older women who reported long-standing regular practice of combined-exercise training at the Center of Sports and Education “Mané Garrincha”, which is part of the Sports Department of São Paulo City, Brazil. Recruited volunteers received information regarding the study before they signed an informed consent that had previously been approved by the Research Ethics Committees of both UNIFESP (approval number 3,623,247) and UNISA (approval number 4,350,476). The present study was carried out in accordance with the Declaration of Helsinki [27].

The same geriatric physician was responsible for clinical and physical examinations. None of the older women who participated in this study presented with HIV and/or other chronic infections; neoplasia, liver, neurological, and/or renal diseases; type 1 diabetes mellitus (exclusion criterion (1)); reported the use of anti-inflammatory drugs; were taking multivitamin/antioxidant supplements (exclusion criterion (2)) in the 2 months prior to data collection (exclusion criterion (3)); or were infected by the SARS-CoV-2 virus until the end the study (exclusion criterion (4)). Furthermore, all volunteers were engaged in the same combined-exercise training program supervised by the same physical education professional (exclusion criterion (5).

### 2.2. Combined-Exercise Training Program

Details regarding the combined-exercise training program performed by the older women volunteers can be found in our previous reports [9,28]. Briefly, this program comprised of a combination of resistance and aerobic exercises performed for 60–75 min per session at moderate intensity 3 times per week on alternate days for 30 days; furthermore, the same physical education professional was always responsible for supervising them.

### 2.3. Anthropometric Data

Body weight data were accurately obtained using a digital scale (Personal^®^ scale, Filizzola, São Paulo, Brazil) to the nearest 0.1 kg, whereas body height data were collected using a wall-mounted stadiometer (accuracy to the nearest 0.1 cm). In order to carry out these evaluations, the older women participants were instructed to wear light clothes and be barefoot. Body mass index (BMI, kg/m^2^) values were calculated based on these data.

### 2.4. Physical Tests and Muscle Strength Assessments

Performance tests were performed according to traditional protocols described in the scientific literature [29,30,31,32]; the results of the TUGT are expressed in seconds (s), whereas GS results are expressed in meters/second (m/s), and HG results are expressed in kilograms of force (kgf). HG measurement was performed with an analog dynamometer (Jamar Hydraulic Hand Dynamometer^®^, Sammons Preston Rolyan, Bollingbrook, IL, USA), with results reflecting the best performance of three attempts with the dominant hand, with an interval of 1 min between each attempt. Appendix A illustrates the physical performance tests applied to the older women who participated in the present study.

The same research assistant, a geriatric physician collaborator in the present study, was responsible for the application of all physical performance tests.

### 2.5. Blood Sample Collection

Blood samples were collected in tubes containing EDTA anticoagulant (ethylenediamine tetra-acetic acid) on two different occasions: before (January–February 2020, pre) and 12 months after (January–February 2021, post) the social isolation imposed in response to the COVID-19 pandemic. Plasma aliquots were obtained after blood tube centrifugation (300× *g*, 10 min at 4 °C) and used to assess the circulating lipid and protein profiles.

### 2.6. Determination of Circulating Lipid and Protein Profiles

Circulating levels of total cholesterol and fractions (LDL and HDL), triglycerides, total protein, albumin, urea, and creatinine were determined using commercial kits (Bioclin-Quibasa, Belo Horizonte, Minas Gerais, Brazil), and the results were analyzed with an automated system (Dimension^®^ RxL Max^®^ Integrated Chemistry System, Siemens, Deerfield, IL, USA). Plasma LDL levels were estimated by the Friedewald formula [33]. Estimated creatinine clearance (ECC) was determined using the Cockcroft–Gault equation [34,35,36].

### 2.7. Statistical Analysis

The normal distribution of data obtained in the present study was statistically analyzed by the Shapiro–Wilk test, followed by evaluation of variance homogeneity through the Levene test. Because all data showed a normal distribution (parametric variables), the paired t-test was used to analyze the differences in the results obtained from anthropometric and metabolic parameters, as well as in the performance on physical tests. Pearson’s rank correlation coefficient was applied to verify the associations between all parameters assessed in this study. The significance level was set to 5% (*p* < 0.05).

## 3. Results

Table 1 shows the anthropometric data from the participant group enrolled in the present study obtained before (pre) and 12 months after (post) the social isolation imposed in response to the COVID-19 pandemic. Both parameters were unchanged during this period.

Figure 1 shows the systemic lipid and protein profiles of volunteers pre- and post-COVID-19 pandemic period assessed in this study. We observed a significant increase in the circulating levels of triglycerides (*p* = 0.0164, Figure 1D) and creatinine (*p* = 0.0152, Figure 1I), as well as a significant decrease in estimated creatinine clearance (*p* = 0.0089, Figure 1J), one year after COVID-19-related social isolation as compared to the values observed pre-COVID-19 pandemic. Although statistical analysis did not show a significant difference between the values of triglycerides and HDL ratio (TG/HDL ratio, Figure 1E), the p-value obtained in this evaluation (*p* = 0.06) showed a remarkable tendency toward enhancement of this ratio during the COVID-19 pandemic period. No other significant differences were found.

Table 2 shows the results concerning physical performance in the functional physical tests (GS and TUGT) and the muscle strength measure (HG) among participants obtained before (pre) and one year after the social isolation imposed in response to the COVID-19 pandemic (post). Higher values of GS and TUGT and lower values of HG were found one year after (post) the pandemic relative to those observed pre-COVID-19 pandemic period. In addition, the results obtained with respect to these parameters in study participants were above the cutoffs for GS (≤0.8 m/s), TUGT (≥20 s), and HG (<16 kg) proposed by the European Working Group on Sarcopenia in Older People (EWGSOP) in a 2019 revision [13], thus demonstrating a low risk of developing sarcopenia.

Table 3 shows only the significant results obtained in the Pearson’s coefficient correlation analysis both before (pre) and one year after (post) the social isolation imposed in response to the COVID-19 pandemic among volunteers who participated in the present study. Several positive and negative correlations were observed at both pre- and post-COVID-19 pandemic time points. However, most significant correlations observed at the pre-COVID-19 pandemic time point were no longer present post-COVID-19 pandemic.

Table 4 shows the results of the multivariate regression analysis adjusted for age. Age had a significant effect on creatinine levels, TG/HDL ratio, and ECC before the COVID-19 pandemic, whereas one year after social isolation, age showed a significant effect on values of TUGT, as well as creatinine levels, TG/HDL ratio, and ECC. No other significant effect was observed.

## 4. Discussion

In this study, we demonstrated that in terms of metabolic parameters, one year of social isolation imposed in response to the COVID-19 pandemic resulted in a significant increase in triglyceride levels and circulating creatinine levels, which can be associated with a significant reduction in creatinine clearance. With respect to functional physical capacity, the time required to perform both GS and TUGT increased, whereas muscle strength was lower post-pandemic than pre-pandemic, although these results were above to the cutoff values related to the development of sarcopenia. Age-adjusted results revealed a considerable impact of age on the assessed metabolic and physical test parameters after one year of social isolation. In addition, most of the correlation results observed pre-COVID-19 pandemic were not observed after one year of social isolation among participants enrolled in this study.

Regarding the alterations observed in the circulating lipid profile in our volunteer group, the increment of triglyceride levels one year after social isolation imposed in response to the COVID-19 pandemic is in agreement with the literature, which reports that every 2.8 h spent practicing sedentary behaviors, especially television watching, is associated with a 0.26 mmol/L increase in triglyceride levels [37]. Corroborating these findings, it has also been reported that sedentary behavior represents a propitious condition for the development significant adverse alterations in metabolic parameters, including triglyceride levels [38]. It is the consensus that the regular exercise training can benefit practitioners in many ways, particularly by modulating/regulating the systemic levels of important metabolic molecules, such as the lipid profile. In this respect, exercise training, including combined-exercise training, can favor a decrease in the circulating levels of total cholesterol, LDL-C, and triglycerides, in contrast to the increment of HDL-C levels [39,40]. Our group previously demonstrated that older women who reported long-standing (more than 18 months) regular practice of a combined-exercise training program presented with lower triglyceride levels than a group of sedentary older women [9,10]. We also reported that according to in vitro assays, the ability of lipid transfer to HDL-C was higher in the group of older women who regularly practiced physical exercises than in the sedentary group, which can decrease atherosclerosis development and progression [10]. Therefore, it is clear that the cessation of regular practice of exercise training for one year during the COVID-19 pandemic negatively altered the metabolic status of older women, particularly in association with a significant elevation of triglyceride levels.

Although HDL-C levels were unchanged during the periods assessed in this study, we evidenced not only an increasing tendency with respect to the triglycerides/HDL ratio (*p* = 0.07) but also a negative significant correlation between circulating levels of triglycerides and HDL-C 12 months after the beginning of the COVID-19 pandemic. According to the literature, the triglycerides/HDL ratio is widely used to predict cardiovascular risk and insulin resistance, which consequently leads to the development of T2DM, including in female patients [41,42]. In this respect, elevation in the systemic levels of triglycerides or reductions in HDL-C, which are related to the occurrence of dyslipidemia, are considered a corollary link between the etiology and pathogenesis of T2DM and the increased risk of atherosclerosis development [41,43]. Our results show that one year of social isolation and interruption of physical exercise training putatively increased the risk of developing both T2DM and cardiovascular diseases among the older women enrolled in this study, reinforcing the importance of regular exercise training to avoid or even minimize the development and progression of these diseases by reducing the triglyceride/HDL-C ratio, including in the older adult population [44].

Beyond these data, we also verified that during the pre-COVID-19 pandemic period, HDL-C levels were negatively correlated with the circulating levels of urea and creatinine. With respect to creatinine, a metabolite produced continuously by the non-enzymatic conversion of creatine and phosphocreatine in muscles, it has been speculated that their serum levels could reflect muscle mass volume, which led to a proposition that serum creatinine levels could also be used as an important indicator for cardiovascular diseases [45], even in individuals without metabolic disorders [46] and T2DM [47]. Regarding the negative correlation observed between the levels of HDL-C and urea before the COVID-19 pandemic, similarly to creatinine, serum urea levels have been shown to be significantly associated with the severity of cardiovascular diseases [47], as their levels are among the major determinants of serum creatinine [48]. This information corroborates our findings concerning the positive correlation between circulating levels of urea and creatinine. This information, as well as the fact that increased circulating HDL-C levels are closely associated with a reduced risk of T2DM and cardiovascular diseases [41], suggests that the negative correlations observed between the systemic levels of HDL-C and creatinine or urea only before the COVID-19 pandemic are related to the regular practice of exercise training, given that 12 months after the COVID-19 pandemic, this significant correlation was no longer observed.

We also observed significant negative correlations between circulating urea levels and triglycerides or the TG/HDL ratio in the post-COVID-19 pandemic period. Individuals with depressive disorders showed not only elevated triglyceride levels but also a decrease in urea levels [49]. Although we cannot confirm, this finding suggests that some volunteers enrolled in the present study may have developed depressive symptoms during the social isolation period imposed in response to the COVID-19 pandemic; for example, Sepúlveda-Loyola et al. reported a significant association between vulnerability and depressive symptoms during the social isolation imposed in response to the COVID-19 pandemic [26].

Corroborating the idea that social isolation in response to the COVID-19 pandemic may have altered several aspects of the population in general, we also verified a significant positive correlation between circulating urea levels and body weight 12 months after the beginning of the pandemic. According to the literature, during the social isolation period imposed in response to the COVID-19 pandemic, remarkable alterations occurred both in terms of food habits and healthy behaviors, which are associated with increased body weight [50,51]. Both alterations in body weight and circulating urea levels are considered to predictors of worse outcomes in COVID-19 patients, particularly in the older adult population [50,51,52]. Therefore, although we did not observe significant alterations in body weight and urea levels between the pre- and post-COVID-19 pandemic period s, for the first time, we demonstrated that two important risk factors for COVID-19 are positively correlated in a group of older women who interrupted their regular practice of exercise training during the pandemic.

Our results suggest a negative association between urea levels and HG values pre-COVID-19 pandemic among our volunteer group. This finding both corroborates the literature [50] and reinforces the notion that regular practice of combined-exercise training is an important factor that can mitigate muscle catabolism, as it was reported that a reduction in muscle mass could enhance urea levels in response to protein breakdown [53] and that reduced muscle mass affects the muscle strength measure in the HG test, which increases the risk of developing sarcopenia in the older adult population [13].

Other important findings obtained in this study that reinforce the benefits of regular practice of exercise training in the older adult population include the correlation analyses between circulating levels of albumin and total protein, both serum and estimated clearance of creatinine, GS, and TUGT. The positive correlation between the levels of albumin and total protein observed before the COVID-19 pandemic was expected, as albumin is the most abundant circulating protein. Systemic albumin levels are not only used as a prominent nutritional biomarker, mainly in aged people [54], but increased blood albumin levels were previously reported in the older adult population who did not present with sarcopenia and frailty as compared to those who presented with these conditions [55,56]. This information corroborates our findings that circulating albumin levels were negatively correlated with the values of GS and TUGT, demonstrating that increased albumin levels resulted in improved performance in physical tests. Furthermore, the negative correlation between systemic levels of albumin and creatinine, in association with the positive correlation between albumin levels and estimated creatinine clearance, corroborates the proposal that albumin is a biomarker of healthy aging, given that, as previously mentioned, increased serum creatinine levels can indicate not only the development of cardiovascular diseases [45] but also renal disruptions. Thus, the positive correlation between albumin levels and creatinine clearance shows that regular practice of a combined-exercise training program can contribute to the maintenance of renal functions. Accordingly, the participants in this study significantly benefited from exercise training, although all such prominent benefits were lost after one year of social isolation imposed in response to the COVID-19 pandemic.

Regarding the significantly increased circulating creatinine levels, in contrast to the decreased estimated creatinine clearance, both can serve as biomarkers of renal function and skeletal muscle mass in healthy populations [51,57]. As previously mentioned, creatinine is a product of creatine metabolism in muscles, and it is eliminated by free filtration in the kidneys. The estimated creatine clearance calculated by the Crockcroft–Gault equation is a simple and quick way to assess the glomerular filtration rate in geriatric clinical practice [58,59]. In older adults, due to the physiological reduction in renal glomerular filtration rate, isolated serum creatinine may underestimate the loss of renal function and overestimate total skeletal muscle mass, which is also reduced in association with the physiological process of aging [60]. Despite the significant increase in mean values of circulating creatinine after one year of the COVID-19 pandemic, the reported levels were within the acceptable range for healthy older adults. Similarly, the mean values of estimated creatinine clearance observed post-COVID-19 pandemic, although lower than those reported pre-COVID-19 period, were compatible with a group of healthy older adults [61]. However, these significant findings suggest remarkable physiological alterations both in muscle status and the glomerular filtration rate associated with interruption of regular practice of exercise training in older women [51].

The positive correlation between circulating creatinine levels and GS observed during the pre-COVID-19 pandemic period corroborates the idea that increased muscle mass (estimated by serum creatinine levels) is associated with faster performance on a 4 m course, also demonstrating that one year of interruption of regular practice of exercise training as a result of the COVID-19 pandemic altered this association.

Whereas during the pre-COVID-19 pandemic period, the estimated creatinine clearance did not prove to be an adequate biomarker of physical performance in our volunteer group, after 12 months of the COVID-19 pandemic period, we observed a positive correlation between estimated glomerular filtration rate and TUGT, a physical test that requires balance, strength, and speed in displacement [30]. This finding reveals that the higher the estimated creatinine clearance value, the longer required to perform the test, indicating worse physical performance. Our results differ from those reported in the literature showing that improved renal function is associated with improved physical performance [57], as the glomerular filtration rate is directly associated with muscle mass and performance.

In order to understand these results, it is worth clarifying that the variables used in the equation (circulating creatinine levels, age, and body weight) to calculate estimated creatine clearance (Cockcroft-Gault equation) independently interfered with TUGT performance. Although we did not observe a significant association between circulating creatinine levels and this functional physical test, body weight values were positively correlated with TUGT, demonstrating that heavier volunteers performed worse on the TUGT, both pre- and post-COVID-19-pandemic period. Similar results have been described in the literature [11,62], demonstrating an association between worse strength and balance performance of lower limbs in older adults and increased weight and a higher percentage of body fat. In agreement with the literature, these associations are the result of the negative influence of adipose tissue on the reduction in neuromuscular function in the older adult population [63,64].

Taken together, our results show that body weight and BMI were better markers of physical performance than estimated creatine clearance among our elderly volunteers both pre- and post-COVID-19 pandemic, especially in terms of TUGT. Our results highlight the advantage of anthropometric measurements (weight and BMI) to assess physical performance, particularly after one year of interruption of regular practice of exercise training, as we observed positive associations between these measurements and HG measurements. According to the literature, the occurrence of such correlations in the detraining phase is probably a result of ceiling effect of these submaximal tests when performed in a group of long-standing regular training practitioners [51].

Specifically in relation to the physical tests, one of the main results obtained in this study is related to the reduction in HG values in association with increased GS and TUGT values after one year of interruption of exercise training as compared to the data obtained pre-COVID-19 pandemic, although these results were maintained above the cutoffs associated with the development of sarcopenia, as proposed by the EWGSOP [13]. Our results are in accordance with the literature, in which social isolation has been significantly associated with adverse physical outcomes in older adults [65,66,67,68].

Multivariate regression analysis adjusted for age showed significant impacts on some metabolic parameters during the pre-pandemic period. However, when the same analysis was performed on data obtained one year after the COVID-19 pandemic, another remarkable impact was observed, particularly in terms of TUGT, which can corroborate our observation that social isolation negatively impacted the physical performance of older adults.

Finally, it is important to cite some limitations of the present study, such as (a) the lack of assessment of body composition among participants; (b) the use of serum creatine and estimated creatine clearance as muscle biomarkers instead of the creatinine dilution test after oral deuterium-labeled creatine ingestion, which is more suitable for estimating muscle mass [13]; (c) the lack of clinical diagnoses in this group; and (d) the impossibility of comparing the data obtained in this study with data of other groups, such as sedentary older women, or aged men and young women presenting with sedentary or active lifestyles. Such information could help us better understand the associations between physical tests and metabolic parameters, especially by demonstrating whether social isolation could similarly affect other populations, as reported here in a group of older women volunteers. Moreover, it is reasonable to suggest that an increase in body fat percentage may have occurred after one year of social isolation imposed in response to the COVID-19 pandemic either due to interruption of exercise training or due to the aging process [51,69,70], although the mean values of body weight and BMI were unchanged in the volunteers during this period.

## 5. Conclusions

Based on the results obtained in the present study, we can conclude that one year of social isolation imposed in response to the COVID-19 pandemic, which interrupted the long-standing regular practice of combined-exercise training by a group of older women, negatively impacted this group, leading to an increased risk of developing syndromes and diseases of aging represented by alterations in some systemic metabolic parameters and worse performance in physical tests of strength, mobility, and balance.

## Figures and Tables

**Figure 1 healthcare-10-01736-f001:**
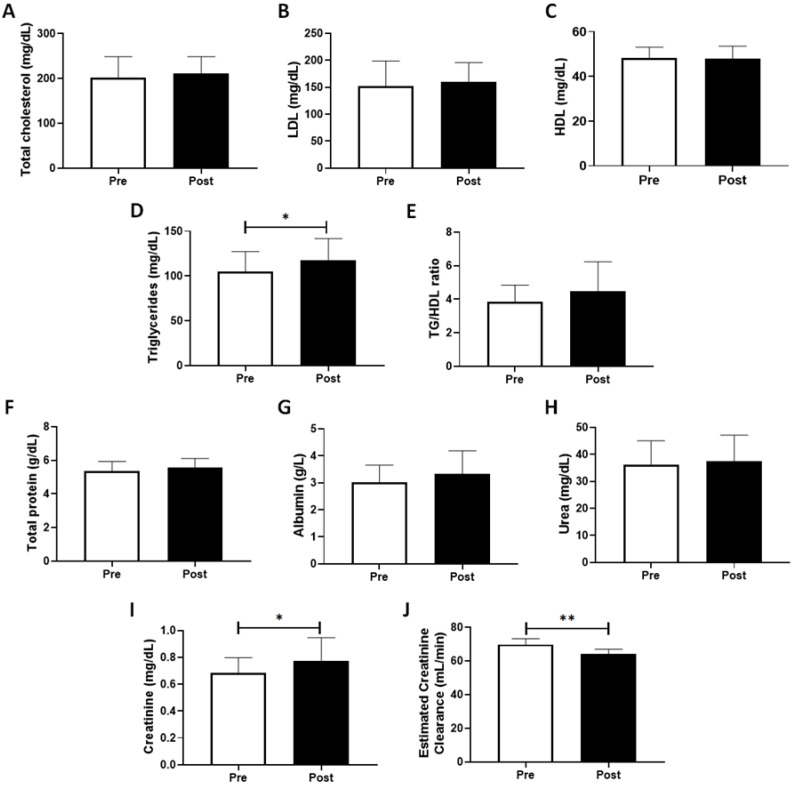
Circulating levels of lipid (total cholesterol (**A**), LDL (**B**), HDL (**C**), triglycerides (**D**) and TG/HDL ratio (**E**)) and protein (total protein (**F**), albumin (**G**), urea (**H**), and creatinine (**I**)) profiles, as well as estimated creatinine clearance (**J**), mean, and standard deviation (SD) of participants before (pre) and after (post) one year of social isolation imposed in response to the COVID-19 pandemic. * *p* < 0.05; and ** *p* < 0.01.

**Table 1 healthcare-10-01736-t001:** Anthropometric data (age, weight, height, and BMI), mean, and standard deviation (SD) of older women volunteers participating in the study.

Variable	Older Women (n = 30)		
	Before (Pre)	After (Post)	*p-*Value
Age (year)	73.7 ± 6.2	74.7 ± 6.2	ns
Weight (kg)	60.9 ± 12.9	61.1 ± 13.1	ns
Height (m)	1.53 ± 0.07	1.53 ± 0.07	ns
BMI (kg/m^2^)	25.8 ± 4.4	25.9 ± 4.6	ns

ns = not significant.

**Table 2 healthcare-10-01736-t002:** Functional physical tests (GS, in meters per seconds (m/s); and TUGT, in seconds (s)) and muscle strength measure (HG, in kilograms of force (kgf)), presented as mean and standard deviation (SD), of older women volunteers participating in the study.

Variable	Older Women (n = 30)		
	Before (Pre)	After (Post)	*p-*Value
GS, m/s	3.2 ± 0.5	4.2 ± 1.2	0.0129
TUGT, s	6.8 ± 0.8	7.6 ± 1.1	0.0012
HG, in kgf	23.5 ± 3.7	21.9 ± 3.9	0.0151

**Table 3 healthcare-10-01736-t003:** Significant results obtained in the Pearson’s coefficient correlation analysis among all parameters assessed both before (pre) and 12 months after (post) social isolation imposed in response to the COVID-19 pandemic. Bold text indicates parameters assessed both pre- and post-COVID-19 pandemic period.

Parameters	Pre		Parameters	Post	
Anthropometric X Metabolic	*rho-*value	*p-*value	Anthropometric X Metabolic	*rho-*value	*p-*value
**Body weight X BMI**	0.909	<0.0001	**Body weight X BMI**	0.917	<0.0001
**Body weight X ECC**	0.595	0.0007	**Body weight X ECC**	0.614	0.0003
**BMI X ECC**	0.467	0.0106	**BMI X ECC**	0.540	0,0021
**Total cholesterol X LDL**	0.988	<0.0001	**Total cholesterol X LDL**	0.991	<0.0001
**Creatinine X ECC**	−0.594	0.0005	**Creatinine X ECC**	−0.479	0.0073
Creatinine X Urea	0.692	<0.0001	Body weight X Urea	0.360	0.0491
Creatinine X HDL	−0.362	0.0489	HDL X Triglycerides	−0.551	0.0016
Creatinine X Albumin	−0.371	0.0432	Urea X Triglycerides	−0.430	0.0175
Albumin X Total protein	0.376	0.0401	Urea X TG/HDL ratio	−0.413	0.0220
Albumin X ECC	0.397	0.0295			
HDL X Urea	−0.430	0.0175			
Anthropometric X Metabolic X Physical test	*rho-*value	*p-*value	Anthropometric X Metabolic X Physical test	*rho-*value	*p-*value
**Body weight X TUGT**	0.581	0.0009	**Body weight X TUGT**	0.469	0.0136
**BMI X TUGT**	0.536	0.0027	**BMI X TUGT**	0.566	0.0020
Albumin X GS	−0.411	0.0237	Body weight X HG	0.390	0.0328
Albumin X TUGT	−0.397	0.0298	BMI X HG	0.433	0.0167
Creatinine X GS	0,.13	0.0231	ECC X TUGT	0.433	0.0239
Urea X HG	−0.406	0.0286			
Physical tests	*rho-*value	*p-*value	Physical tests	*rho-*value	*p-*value
**GS X TUGT**	0.386	0.0346	**GS X TUGT**	0.451	0.0180

BMI = body mass index; ECC = estimated creatinine clearance; GS = gait speed; HDL = high-density lipoprotein; HG = handgrip; LDL = low-density lipoprotein; TUGT = time-up-and-go test.

**Table 4 healthcare-10-01736-t004:** Results of multivariate regression analysis adjusted for age.

Variables	Age-Adjusted							
	Older Women (n = 30)							
	Pre-COVID-19 Pandemic				Post-COVID-19 Pandemic			
	*β**−*Value	95% CI	*p-*Value	*R^2^*	*β**−*Value	95% CI	*p-*Value	*R^2^*
TG/HDL ratio	0.009	0.002 to 0.017	0.0163	0.834	0.014	0.0014 to 0.026	0.032	0.906
Creatinine (mg/dL)	−0.011	−0.017 to −0.0045	0.003	0.785	−0.0128	−0.019 to −0.007	0.0006	0.815
ECC (mL/min)	−1.139	−1.499 to −0.780	<0.0001	0.517	−0.9887	−1.310 to −0.667	<0.0001	0.686
TUGT (s)	ns	ns	ns	Ns	0.2432	0.104 to 0.383	0.003	0.858

## Data Availability

Data availability the responsibility of the authors, and any data can be made available upon reasonable request.

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
