# Peer review of "Older Women Who Practiced Physical Exercises before the COVID-19 Pandemic Present Metabolic Alterations and Worsened Functional Physical Capacity after One Year of Social Isolation"

_healthcare, 2022, doi:10.3390/healthcare10091736_

Round 1

Reviewer 1 Report

 In this research, the authors investigated the effects of one year of social isolation imposed by COVID-19 on the metabolic parameters and functional physical capacity of older women. Because these women regularly practiced physical exercises before the pandemic, some systemic metabolic parameters were altered, and estimated creatinine clearance and functional physical capacity became worse. The research is useful for understanding the impact of COVID-19 on the health of older women. However, there was some issue need to be considered.

1. In this paper, the authors investigated the systemic metabolic parameters of older women based on combined-exercise training programs, such as used G*Power program, according to their previous research. However, authors still can provide more information, especially the experiment evidence like images, such as muscle strength test pictures, to help readers to understand the experiment processes and how the authors obtain the data.

2. The reviewer thinks the biggest problem of this article is why only women's data have been used. During the COVID-19 impact, nobody goes outside. Then, the results (some systemic metabolic parameters worsened) that the authors suggested are obvious since they cannot do exercises like before COVID-19. If the authors can add old man data, young women, or some other groups to make a comparison, the results will be very interesting and meaningful.

3. Except for discussing systemic metabolic parameters and estimated creatinine clearance and functional physical capacity, giving some metal heath discussions about the results might improve the originality of this research.

Reviewer 2 Report

I would like to thank the authors for the study presented in the manuscript. Overall I find the manuscript well written regarding the materials and methods, results presented, discussion and conclusions.

Myself not being English native speaker, I found some parts of the text difficult to perceive and therefore my doubts mainly are related to the language and terminology used. 

1) The title of the manuscript "Physically-exercised older women present significant metabolic alterations and worsened functional physical capacity after one year of the COVID-19 pandemic" somehow misleads to think that despite performing physical exercises (not clear- before or during the one year of pandemic) older women present significant metabolic alterations and worsened functional physical capacity. Of course, later it is well explained that they have long term experience to participate in combined- exercise training program.  

2) “Physically -exercised older women”- I have not met such expression in scientific literature earlier, usually the term "physical activity" is used. Perhaps, idea is to emphasize that the participants had experience to participate in specific training program which represents the  planned, structured, and repetitive physical activity and has as a final or an intermediate aim the improvement or maintenance of physical fitness (according to Caspersen CJ, Powell KE, Christenson GM. Physical activity, exercise, and physical fitness: definitions and distinctions for health-related research. Public Health Rep.1985;100(2):126–131).

3) “All experiments were carried out in agreement with the Declaration of Helsinki”(Line 131) – the study itself do not have experimental nature, it is observational study. Does this apply simply to the process of data collection (tests, assessments, blood samples collection)?

4) “ sampling collection” (Line 137)- does this related to the blood samples collection or sampling in general (participants recruitment)?

5) “researcher participant” (Line 163)- does it means the researcher or researcher assistant responsible for data collection (physical tests and muscle strength assessments)?

6) In result section several times expression “the group of older women physically-exercised” used (Lines 188-189, 214, 225). Just “participants” would be fine as the sampling procedure is described earlier.

7) The titles for functional physical tests and their abbreviations are described already in Introduction (Lines 76- 78) but later they are used as abbreviations as well as full titles (description of Table  2, Lines 212- 214).

8) The findings on functional physical tests initially are simply described without interpretation that they indicate the worsening of functional physical capacity. The interpretation appears only in last paragraph of Discussion (Lines 412- 417).

9) Some limitations of the study is described at the end of Discussion. I would expect that there would be interest of researchers to perform comparative study of two groups of older women – one group of participants with experience of regular physical activity and another group of participants without. Participants in the study was women over 70 yrs and the effect of changes related to ageing also could be considered, especially if more detailed information on their health status was not available.

Round 2

Reviewer 1 Report

The authors have responded to the reviews.